

# Biodiversity assessment and environmental risk analysis of the single line transgenic pod borer resistant cowpea

Abraham Isah[1,2], Rebeccah Wusa Ndana[1], Yoila David Malann[1], Onyekachi Francis Nwankwo[3], Abdulrazak Baba Ibrahim[4] and Rose Suniso Gidado[2,5]

[1] Department of Biological Sciences, Faculty of Science, University of Abuja, Abuja, FCT, Nigeria
[2] Open Forum on Agricultural Biotechnology in Africa, Nigeria Chapter, National Biotechnology Development Agency, Abuja, FCT, Nigeria
[3] Product Stewardship, African Agricultural Technology Foundation, ILRI Campus, Nairobi, Kenya
[4] Forum for Agricultural Research in Africa, New Achimota Mile 7, Accra, Ghana
[5] Department of Agricultural Biotechnology, National Biotechnology Development Agency, Abuja, FCT, Nigeria

Corresponding author
Abraham Isah,
abraham2637@gmail.com

## ABSTRACT

**Background**. The discussion surrounding biological diversity has reached a critical point with the introduction of Nigeria's first transgenic food crop, the pod borer-resistant (PBR) cowpea. Questions have been raised about the potential risks of the transgenic *Maruca vitrata*-resistant cowpea to human health and beneficial insects. Public apprehension, coupled with social activists' calling for the removal of this crop from the nation's food market, persists. However, there is a lack of data to counter the assertion that cultivating PBR cowpea may have adverse effects on biodiversity and the overall ecological system. This research, with its multifaceted objective of examining the environmental safety of PBR cowpea and assessing its impact on biodiversity compared to its non-transgenic counterpart, IT97KN, is of utmost importance in providing the necessary data to address these concerns.

**Methods**. Seeds for both the transgenic PBR cowpea and its isoline were obtained from the Institute for Agricultural Research (IAR) Zaria before planting at various farm sites (*Addae et al., 2020*). Throughout the experiment, local cultural practices were strictly followed to cultivate both transgenic and non-transgenic cowpeas. Elaborate taxonomic keys were used to identify arthropods and other non-targeted organisms. Principal component analysis (PCA) was used to evaluate potential modifications in all ecological niches of the crops. The lmer function of the R package lme4 was used to analyze diversity indices, including Shannon, Pielou, and Simpson. The Bray–Curtis index was used to analyzed potential modifications in the dissimilarities of non-targeted organisms' communities.

**Results**. Examination of ecological species abundance per counting week (CW) revealed no disruption in the biological properties of non-targeted species due to the cultivation of transgenic PBR cowpea. Analysis of species evenness and diversity indices indicated no significant difference between the fields of transgenic PBR cowpea and its isoline. Principal component analysis results demonstrated that planting PBR cowpea did not create an imbalance in the distribution of ecological species. All species and families observed during this study were more abundant in transgenic PBR cowpea fields than in non-transgenic cowpea fields, suggesting that the transformation of cowpea does not

negatively impact non-targeted organisms and their communities. Evolution dynamics of the species community between transgenic and non-transgenic cowpea fields showed a similar trend throughout the study period, with no significant divergence induced in the community structure because of PBR cowpea planting. This study concludes that planting transgenic PBR cowpea positively influences biodiversity and the environment.

## INTRODUCTION

Researchers coined the term biodiversity from the word biological diversity to refer to the heterogeneity and variability of the total number of biological organisms found within a given habitat or ecosystem at any given time (*Roe, Seddon & Elliott, 2019*; *Adom et al., 2019*; *Meine, 2018*; *Rawat & Agarwal, 2015*). The concept of biodiversity is multidimensional, encompassing genetics, species, and ecology. Several studies, including *Tilman, Isbell & Cowles (2014)* and *Malhi et al. (2020)*, have revealed that the degree of variability of living organisms on earth plays a crucial role in sustaining the ecosystem and could serve as a major indicator for predicting the safety of any environment at any given time. The productivity and efficiency of any agricultural system around the world can be strongly influenced by its varietal and species diversity over an extensive scale of conditions (*Pawlak & Kołodziejczak, 2020*; *Carpenter, 2011*; *Krishna, Zilberman & Qaim, 2009*). Biodiversity also plays a crucial role in enhancing an organism's resilience to stress and shocks, as well as its adaptability to new and challenging environmental conditions. Additionally, it is a vital factor in the sustainability of production systems and genetic improvement (*Vasiliev, 2022*; *Ortiz et al., 2021*). With the negative impact of climate change, characterized by increased crop pest infestation and decreased agricultural soil fertility on a global scale (*Malhi, Kaur & Kaushik, 2021*; *Habib-ur Rahman et al., 2022*; *Subedi, Poudel & Aryal, 2023*), it is crucial to emphasize the importance of sustaining and enhancing the variability of crop and animal genetic resources. This variability is essential for ensuring the resilience and stability of living organisms over time.

After about thirty years of the safe use of transgenic crops with more than 3 million hectares planted across Africa (*Endale et al., 2022*) and their recorded benefits (*Gbadegesin et al., 2022*; *Smyth, 2020*), debate and concerns about their environmental effects have continued to persist (*Gbadegesin et al., 2022*; *Gbashi et al., 2021*; *Smyth et al., 2021*). Critical among the issues discussed so far is its potential impact on biodiversity (*Fernandes et al., 2022*; *Lucht, 2015*). The quest to safeguard the orphan crop, cowpea, often referred to as "poor man's meat" for its vital role as an affordable protein source in third-world countries, from the devastating impact of the *Maruca vitrata* insect pest has led to its transformation using the *Cry1Ab* protein. Derived from the soil bacterium *Bacillus thuringiensis*, *Cry1Ab* selectively targets specific receptors in the digestive systems of susceptible pests, making it a widely utilized biopesticide in agricultural biotechnology, effectively conferring resistance

against certain insect pests such as the pod borer *Maruca vitrata* and reducing reliance on chemical pesticides. Though some studies, including *O'Callaghan et al. (2005)* and *Romeis et al. (2014)*, have suggested that the insecticidal property of the *Cry1Ab* protein may be toxic to non-target species, including herbivores, parasitoids, and predators, many of these studies examined the impact of this protein on species in non-natural systems without taking into account ecological interactions or the actual level of exposure of vulnerable stages in natural settings (*Dale, Clarke & Fontes, 2002*). Conducting additional studies that consider complex systems and exposure conditions akin to those encountered in the field could offer more realistic insights into the potential detrimental effects of *Bacillus thuringiensis* (*Bt*) crops on non-target organisms (*Sears et al., 2001*).

In the guidance documents of the European Food Safety Authority *EFSA (2016)*, conserving biodiversity is emphasized as a major goal in environmental protection, highlighting its magnitude and significance. Quantifying biodiversity is a prerequisite for reaching set targets. Since Nigeria commercialized its first transgenic crop, insect-resistant (IR) cotton, in 2018 and joined the league of biotech countries, it has triggered a general debate in Africa on the potential impact of transgenic crops on biodiversity (*Endale et al., 2022*). The introduction of her first transgenic food crop, pod borer resistant (PBR) cowpea, in 2019, has further exacerbated these concerns among Nigeria's stakeholders. A significant concern in Nigeria regarding the safety of introducing transgenic PBR cowpea revolves around its potential to negatively impact species and ecosystem diversity. Key stakeholders speculate that its toxicity to the targeted insect, *Maruca vitrata*, raises concerns about its impact on non-targeted organisms (NTOs), including those crucial for ecosystem functioning. Currently, there is a paucity of data to refute claims that this transgenic PBR cowpea supports biodiversity and is safe for our environment. This study, therefore, focuses on the biodiversity assessment of the single-line transgenic pod borer-resistant cowpea to evaluate its potential impacts on non-target organisms.

## MATERIALS & METHODS

### PBR cowpea seeds and its isoline

Seeds of both transgenic PBR cowpea (IT97KT) and its isoline, IT97KN, were provided by the Institute for Agricultural Research (IAR) Zaria before planting at various farm sites. The *Cry1Ab* event in the PBR cowpea was confirmed using the lateral flow strip kits obtained from Qiagen Inc. at the Mary Halaway Laboratory, Department of Biochemistry, Faculty of Life Sciences, Ahmadu Bello University: 5 g, each of transgenic and non-transgenic seeds were mashed separately in two different mortars and pestles, after which the extraction buffer was added to each container. The flow strip was then inserted and allowed to stay for about 10 min, after which the lines were read (Fig. S1).

### Experimental design and sampling

The two cowpea lines, IT97KT and IT97KN were planted in three different farms of the National Biotechnology Research and Development Agency (NBRDA) from February to May, August to November 2022, and February to May 2023 using the irrigation farming method during the dry season with three replications on each farm site (Fig. S2). Both

cowpea lines were grown following local cultural practices throughout the experiment. The two crop varieties, transgenic (IT97KT) and non-transgenic isoline cowpea (ITN97KN) were planted in a randomized block design with 3 replications (Fig. S2). The measurement of each plot was estimated at 10 m by 15 m, encompassing eight 30 cm interspaced rows with 25 cm of space between each plant. 3 m of plain boundaries were created to function as seclusion among plots (Fig. S2). No crop was planted on the three research farms one year before the research. In addition, no herbicide or insecticide was used before or during the study period.

## Identification of species to family and to functional groups

Arthropods and other non-targeted organisms were identified by using suitable and elaborated dichotomous taxonomic keys, according to *Goulet & Huber (1993)*, *Triplehorn, Johnson & Borror (2005)*, and *Jenny et al. (2017)*. The taxonomic grouping was done using the family level as default, while in cases where classification based on family level was not obtainable, priority was given to the order and suborder to which the organism belongs (*Jenny et al., 2017*). The individual organisms were further grouped into predator, parasitoid, and herbivore ecological functional groups. Throughout the study period, no organisms were recorded as unknown. The counting of individual organisms across all three sites commenced 21 days after planting and was designated as the counting week (CW).

## Non-target organism community structure

Possible moderations that may have accrued from planting the transgenic PBR cowpea were analyzed using a precise redundancy analysis (RDA) ordination method called the principal component analysis (PCA) (*Vanden-Brink et al., 2009*), as recommended by *Cuppen et al. (2000)* and *Moser et al. (2007)* to be suitable for assessing the impacts of any plants or animals on the ecosystem. The PCA multivariate technique facilitates the understanding of the interaction between the organisms and their environment (*Moser et al., 2007*) by analyzing the possible effects of the transgenic PBR cowpea on the community species and the resulting changes in the community structure throughout the study period.

## Structural dissimilarity analysis

The analysis for the potential modification in the dissimilarities of the non-targeted organisms' communities between the transgenic PBR cowpea (IT97KT) and its non-transgenic isoline (IT97KN) was done using the Bray–Curtis (BC) index. It evaluates the degree of dissimilarity or similarity between two or more samples using a range of zero (similar) to one (dissimilar) (*Krebs, 1989*; *Bray & Curtis, 1957*). The structural dissimilarity analysis was divided into two phases. In the first phase of the analysis, the Bray-Curtis index was computed using the data collected between all the pairs of the sample plots, IT97KT and IT97KN, on each sampling date. Bray–Curtis dissimilarity ranges between 0 and 1, where 0 indicates that the niches have no dissimilarity, while 1 indicates that the two niches have complete dissimilarity (*Ricotta & Podani, 2017*). Similar procedures were repeated for the second phase of the analysis, where data was collected within each cowpea plot (*Collins,*

*Micheli & Hartt, 2000*) and then followed by a computation of the mean abundance for the respective taxonomic group in line IT97KT and IT97KN per sampling date.

The Bray–Curtis dissimilarity was calculated as: $BC_{ij} = 1 - (2*C_{ij})/(S_i + S_j)$

Where $C_{ij} =$ The sum of the lesser values for the species found in each site.

$S_i$: The total number of specimens counted at site i

$S_j$: The total number of specimens counted at site j

The values for the mean abundance were thereafter used to estimate the BC distance between the respective treatment sampling dates. A linear regression analysis of the data obtained from the BC distance estimation was conducted *versus* the time-lag data.

## Statistical analysis

The total number (N) of arthropods on each plot in the three different farm sites was taken per CW and over the entire period of the study and then divided by the total number of farm sites to get the average. All statistical analyses were performed using R version 4.2.0 (*R Core Team, 2022*) and an Excel spreadsheet. The analysis of the diversity indices, including Shannon (H), Pielou (J), and Simpson (D), facilitates a comparative assessment of the community structures between different treatments in the fields (*Boyle et al., 1990*; *Magurran, 2004*; *Pielou, 1966*; *Oksanen, 2013*) using the lmer function of R package lme4 (*Bates, Mächler & Bolker, 2015*) with cowpea variety (Bt or non-Bt) and time (date of sampling) as fixed factors (*Guo et al., 2014*). A comparison of the mean values of all the scoring parameters, including H, D, J, and N, was done using a one-way analysis of variance (ANOVA).

A covariance analysis was used to conduct a comparative study of the slopes of the regression lines of the two treatments. The parasitoid, herbivore, and predator nutritional relationships were used to classify the whole organisms into three guilds according to *Heong, Aquino & Barrion (1991)* and *Zhang et al. (2011)*. The density of the three guilds was analyzed using one-way ANOVA for each cowpea variety and sampling date. The population of various treatments, herbivore, parasitoid, and predator nutritional guild was defined by using the formula $P_i 5 N_i / N$, where the population of the herbivore, parasitoid, and predator was connoted as $N_i$ while the treatment's entire summed abundance was connoted as N. The species count for each community organism in the various guilds was defined by the formula $P_i 5 N_i / N$, where Ni was defined as the summed ith species and N was the guild count in the respective treatment. The rare, common, and dominant groups were denoted by $P_i < 1\%$, $1\% \leq P_i < 10\%$, and $P_i \geq 10\%$, respectively (*Li & Liu, 2013*).

## RESULTS

### Transgene status confirmation of the cowpea samples

The confirmation of the Cry1Ab event expressed in the PBR cowpea shows a positive result, as seen in Fig. S1. Further tests for the presence of the Cry1Ab gene using the event-specific flow strip in the isoline of the PBR cowpea showed negative results, meaning that the isoline is not transgenic (Fig. S1).

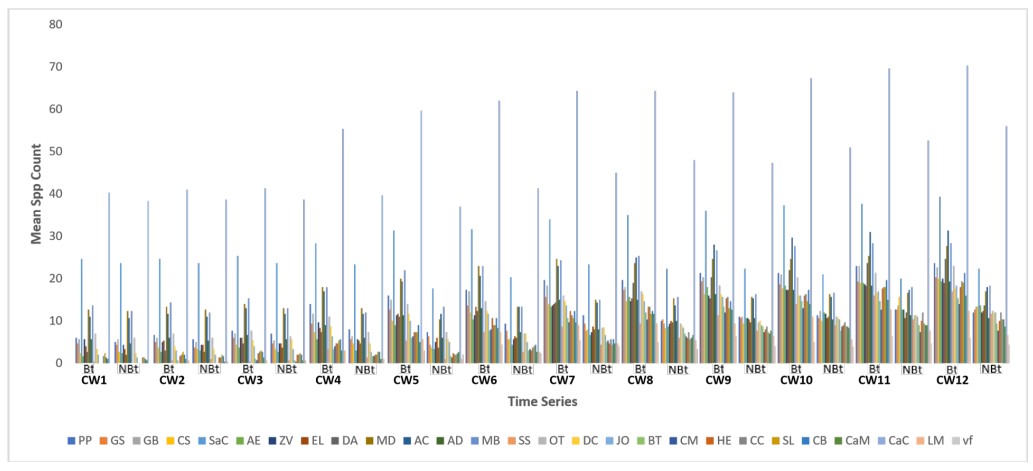

**Figure 1    Mean spp activity overview on field of transgenic and non-transgenic cowpea.**

## Ecological pattern of the transgenic and non-transgenic cowpea field

This study identified the following species in both fields of transgenic cowpea and non-transgenic cowpea: *Pirata piraticus* Clerck, 1757 (pp), *Conozoa hyaline* Forbes, 1848 (GS); *Graphoderus bilineatus* De Geer, 1774 (GB); *Sarcophaga crassipalpis* Macquart, 1850 (SaC); *Alydus eurinus* Say, 1832 (AE); *Zonecerus variegatus* Fabricius, 1775 (ZV); *Romalea microptera* Beauvois, 1817 (EL); *Deudorix antalus* Hopffer, 1855 (DA); *Musca domestica* Linnaeus, 1758 (MD); *Atta cephalotes* Linnaeus, 1758 (AC); *Apis dorsata* Fabricius, 1793 (AD); *Messor barbarus* Linnaeus, 1767 (MB); *Scarabaeus satyrus* Fabricius, 1787 (SS); *Odontoponera transversa,* Smith, 1858 (OT); *Dysdercus cingulatus* Fabricius, 1798 (DC); *Junonia oenone* Linnaeus, 1758 (JO)*; Bombus terrestris* Linnaeus, 1758 (BT); *Chrysomya megacephala* Fabricius, 1794 (CM); *Hypolycaena erylus* Godart, 1824 (HE); *Conozoa carinata* Lamarck, 1816 (CC); *Stenolophus lecontei* Chaudoir, 1869 (SL); *Chorthippus biguttulus* Linnaeus, 1758 (CB); *Carausius morosus* Sinéty, 1901 (Cam); *Camponotus cruentatus* Latreille, 1802 (CaC); *Lilioceris merdigera* Linnaeus, 1758 (LM); *Chilocorus stigma* Say, 1832 (Cst); *Euptoieta claudia* Cramer, 1776  (vf).

The examination of species disparities and distribution indicates no variations between both treatments during CW 1, which  commenced 21 days after planting (Fig. 1 and Table 1). However, from CW 2 to 12, a notable discrepancy was noted in species activities between the transgenic crop field and the non-transgenic cowpea field, with the former exhibiting notably higher species activities.

## Estimated species diversity

From the results of the univariate analyses of the ecological niches of both line IT97KT and line IT97KN, the estimated biodiversity indices (H, J, and D) revealed no significant difference between the two treatments, except during the differentiated flowering time observed between the two cowpea lines (Table 1 and Figs. 2A, 2B, and 2C). The habitat information provided from the Shannon diversity index analysis shows that both habitats

Isah et al. (2024), *PeerJ*, DOI 10.7717/peerj.18094

**Table 1  Statistical parameters of all the mean value analysis of IT97KT vs IT97KN.**

| Wk | Vr | Actual Count | | | Shannon | | | Simpson | | |
|---|---|---|---|---|---|---|---|---|---|---|
| | | Mean | *P* value | *R²value* | Mean | *P* value | *R²* value | Mean | *P* value | R² value |
| 1 | IT97KT | 159.6667 ± 3.6742 | 0.11456 | 0.977 | 2.5258 ± 0.0207 | 0.1690 | 0.881 | 0.8823 ± 0.002 | 0.1006 | 0.976 |
| | IT97KN | 145.6667 ± 3.6742 | | | 2.4641 ± 0.0207 | | | 0.8747 ± 0.002 | | |
| 2 | IT97KT | 173.6667 ± 2.5495* | 0.03418 | 0.988 | 2.6610 ± 0.0134 | 0.0668 | 0.96 | 0.8951 ± 0.002 | 0.0749 | 0.978 |
| | IT97KN | 154.6667 ± 2.5495* | | | 2.5914 ± 0.0134 | | | 0.8853 ± 0.002 | | |
| 3 | IT97KT | 192.6667 ± 2.7183* | 0.02406 | 0.988 | 2.7541 ± 0.0312 | 0.3165 | 0.848 | 0.9067 ± 0.003 | 0.18131 | 0.958 |
| | IT97KN | 168.3333 ± 2.7183* | | | 2.6957 ± 0.0312 | | | 0.8989 ± 0.003 | | |
| 4 | IT97KT | 282.0000 ± 3.4238** | 0.002231 | 0.996 | 2.9239 ± 0.0583 | 0.2144 | 0.693 | 0.9238 ± 0.010 | 0.3374 | 0.562 |
| | IT97KN | 179.6667 ± 3.4238** | | | 2.7760 ± 0.0583 | | | 0.9058 ± 0.010 | | |
| 5 | IT97KT | 358.3333 ± 2.3921*** | 0.0003295 | 0.999 | 3.0512 ± 0.0427 | 0.0665 | 0.889 | 0.9375 ± 0.009 | 0.1345 | 0.806 |
| | IT97KN | 172.0000 ± 2.3921*** | | | 2.8290 ± 0.0427 | | | 0.9082 ± 0.009 | | |
| 6 | IT97KT | 401.6667 ± 9.6724** | 0.005072 | 0.99 | 3.1006 ± 0.0171 | 0.024* | 0.957 | 0.9425 ± 0.004 | 0.0744 | 0.886 |
| | IT97KN | 210.3333 ± 9.6724** | | | 2.9475 ± 0.0171 | | | 0.9222 ± 0.004 | | |
| 7 | IT97KT | 452.3333 ± 8.0312** | 0.003445 | 0.993 | 3.1334 ± 0.0097* | 0.0207 | 0.964 | 0.9464 ± 0.002 | 0.05671 | 0.911 |
| | IT97KN | 259.3333 ± 8.0312** | | | 3.0396 ± 0.0097* | | | 0.9339 ± 0.002 | | |
| 8 | IT97KT | 479 ± 11.1131** | 0.006634 | 0.987 | 3.1506 ± 0.011* | 0.0485 | 0.941 | 0.9485 ± 0.003 | 0.1051 | 0.882 |
| | IT97KN | 287 ± 11.1131** | | | 3.0823 ± 0.011* | | | 0.9385 ± 0.003 | | |
| 9 | IT97KT | 516.6667 ± 8.5765** | 0.003505 | 0.993 | 3.1716 ± 0.0153 | 0.1703 | 0.85 | 0.9510 ± 0.002 | 0.1731 | 0.84 |
| | IT97KN | 312.3333 ± 8.5765** | | | 3.1260 ± 0.0153 | | | 0.9438 ± 0.002 | | |
| 10 | IT97KT | 546 ± 8.9536** | 0.003483 | 0.993 | 3.1761 ± 0.0106 | 0.0919 | 0.914 | 0.9515 ± 0.002 | 0.1445 | 0.856 |
| | IT97KN | 332 ± 8.9536** | | | 3.1303 ± 0.0106 | | | 0.9444 ± 0.002 | | |
| 11 | IT97KT | 580.0000 ± 7.728** | 0.002474 | 0.995 | 3.1859 ± 0.011 | 0.1399 | 0.878 | 0.9515 ± 0.002 | 0.1445 | 0.856 |
| | IT97KN | 360.6667 ± 7.728** | | | 3.1489 ± 0.011 | | | 0.9444 ± 0.002 | | |
| 12 | IT97KT | 603.3333 ± 4.7317*** | 0.0008405 | 0.998 | 3.1913 ± 0.0104 | 0.2600 | 0.902 | 0.9532 ± 0.002 | 0.1003 | 0.871 |
| | IT97KN | 372.6667 ± 4.7317*** | | | 3.1388 ± 0.0104 | | | 0.9458 ± 0.002 | | |

**Notes.**

Vr, variety; Wk, Week.

*Statistically significant.

**Higher level of statistical significance.

***Stronger level of statistical significance.

dominated by the transgenic and non-transgenic cowpea have high species richness and evenness throughout the CWs. Results obtained from the analysis using the Shannon diversity index revealed a close-range value between the transgenic and non-transgenic cowpea habitats. A higher Shannon score was observed for transgenic cowpea fields within the counting weeks of 3 to 8, where flowering was peak. The diversity index score for IT97KN went slightly higher during the counting weeks when its flowering was also at its peak. Results from the analysis of variance show no significant difference at weeks 1, 2, 9, 10, 11, and 12 against the subsequent counting weeks of 4, 5, 6, and 7 (Fig. 2A). Analysis of the Simpson diversity indices shows similar trends in both transgenic and non-transgenic cowpea fields, with both fields recording their lowest Simpson score at CW 1 and 2, respectively. Figure 2B shows that the highest Simpson scores were observed during CWs 11 and 12 in both transgenic and non-transgenic cowpea fields. Analysis of the Pielou Evenness Index shows that the distribution of the individual species is even across the habitat of transgenic and non-transgenic cowpea (Fig. 2C). Further analysis using the regression line plot between the ecological niches of transgenic and non-transgenic cowpea shows a strong positive correlation with a $p$ and $r$ value of 1.810599e−06 and 0.9522146, respectively (Fig. 3A). As the number of species in the ecological niches of PBR cowpea increases, the number of species in its non-transgenic isoline, IT97KN, also increases (Fig. 3A).

Similar results were observed when the ecological niches of transgenic cowpea (IT97KT) and its non-transgenic isoline (IT97KN) were correlated with time (Fig. 3B). The $p$ and $r$ values of 3.42862e−09 and 0.9865187, respectively, were observed for transgenic cowpea $vs$ time, while $p$ and $r$ values of 1.535e−07 and 0.9522146, respectively, were observed for non-transgenic cowpea $vs$ time (Table 2).

## Analysis of the evolution dynamics of the transgenic and non-transgenic cowpea

### Component analysis

Analysis using the multivariate principal component technique reveals no significant differences in the ecological composition of the entire study fields throughout the counting weeks (Figs. 4A and 4B). The essence of the PCA output is to give a clear interpretation of the species points with similar composition—the species scores, which are represented by arrows, point in the direction of increasing abundance. The angle size between a species arrow and another species arrow is inversely correlated, meaning that the smaller the angle size between two species arrows, the stronger the correlation, and the reverse means a weaker correlation within the space. The result output shows a strong positive correlation between EI and DC in both transgenic and non-transgenic cowpea fields. The formation of a right angle between two species arrows means no correlation, while the formation of an opposite angle means a strong negative correlation (*Bioturing, 2018*). The PCA output also attributes significant value to the direction of the species arrow regarding its angle with the principal component axes within the space. The PC analysis from this study shows that AC and Cs strongly influence PC1, while PP and Zv strongly influence PC2, having a heavier weight in the transgenic cowpea field. Md and SaC are the most heavily weighted in PC1,

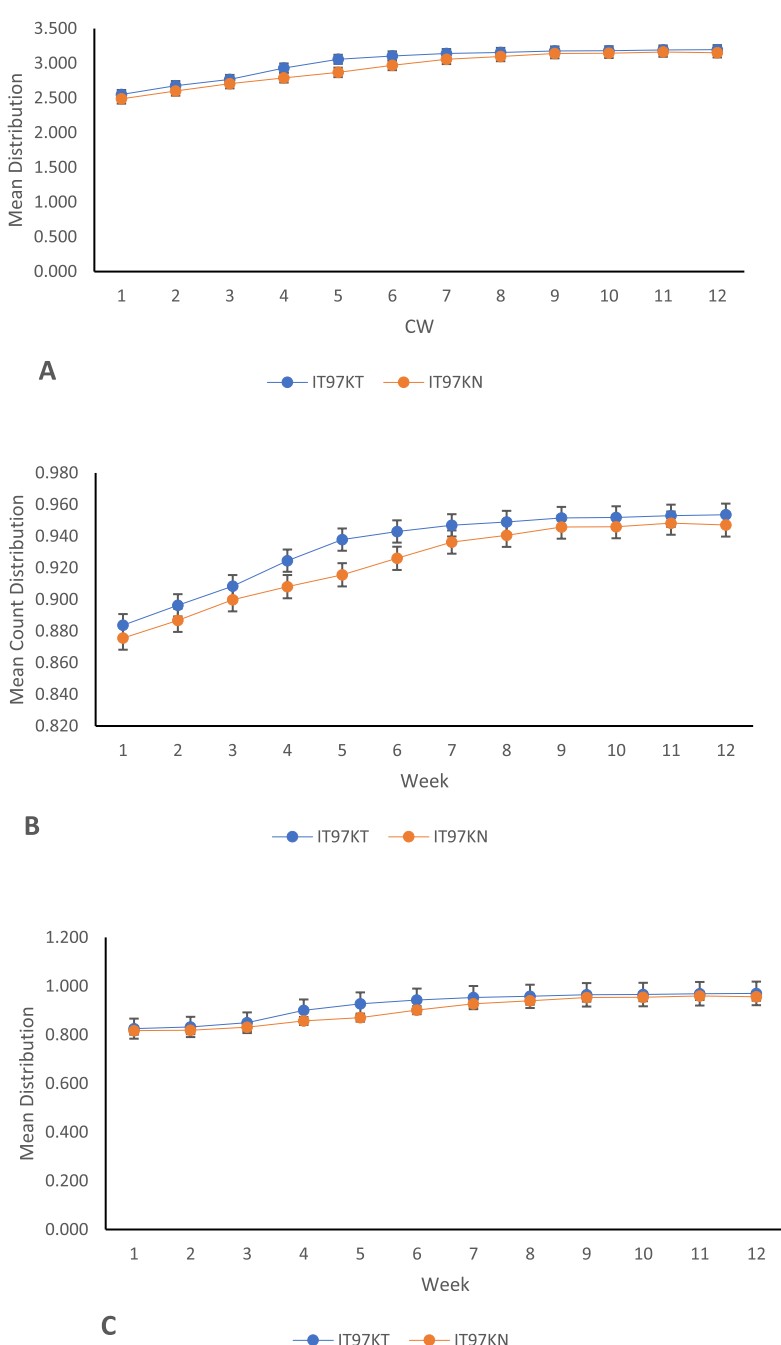

**Figure 2** **Mean line trend analysis of IT97KT (transgenic) *vs* IT97KN (non-transgenic) cowpea in a 12-week spread count using: (A) Shannon; (B) Simpson; (InvSimpson) and (C) Pielou.**

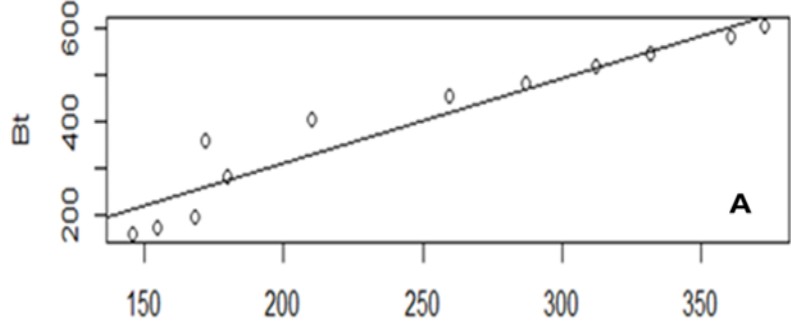

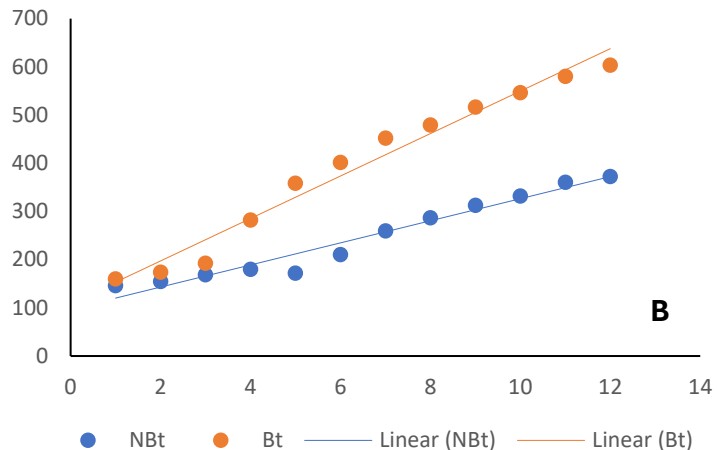

**Figure 3  Line graph.**

**Table 2  Correlation analysis of Bt *vs* NBt, Bt *vs* time (weeks) and NBt *vs* time.**

| Parameters | *p*-value | *r* |
|---|---|---|
| *Bt vs NBt* | *1.535e−07****** | 0.9522146 |
| *Bt vs Time* | *3.42862e−09* | *0.9865187* |
| NBt vs Time (week) | *3.508742e−08* | *0.9784767* |

**Notes.**
*Statistically significant.
**Higher level of statistical significance.
***Stronger level of statistical significance.

strongly influencing the PC1 of the non-transgenic cowpea, while GB and PP are the most heavily weighted species of PC2 in the non-transgenic cowpea field.

The estimation of the number of statistically significant principal components for the ecological niches of both transgenic and non-transgenic cowpea is presented in Fig. 5 below. The number of breakpoints (10) distribution is similar for both ecological niches.

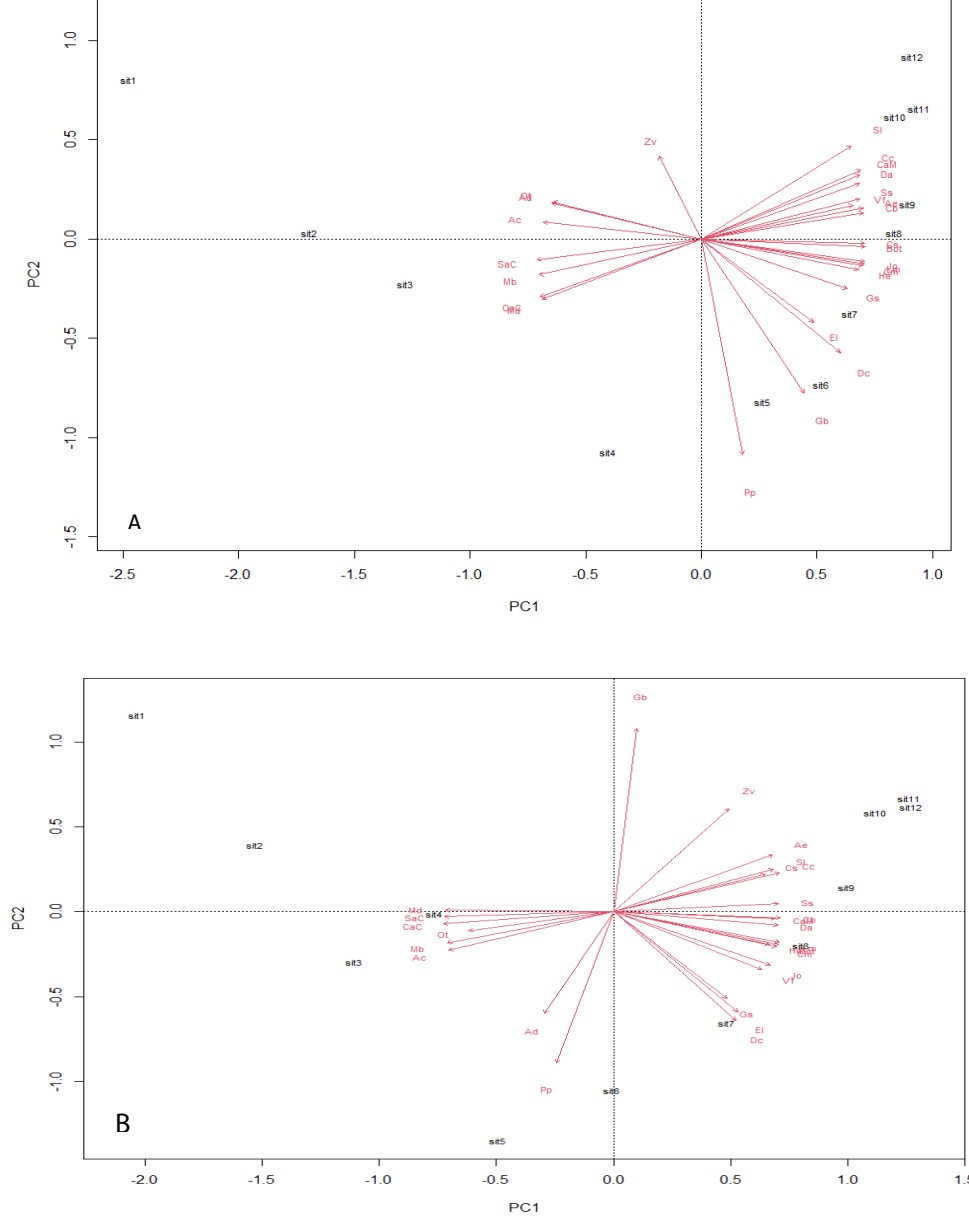

**Figure 4** **Principal component plot analysis.** (A) Bt, PCA Plot; (B) NBt, PCA Plot.

## Composition of organism community of both the transgenic and non-transgenic species

As shown in the figure below, three major guilds, herbivores, parasitoids, and predators were identified throughout the study period (Figs. 6A–6C). The guild analysis for both the transgenic (IT97KT) and non-transgenic (IT97KN) fields reveals the identification of 12, different species in the herbivore, parasitoid, and predator guild. Most species in both fields are herbivores, while the predatory guild has the least number of organisms.

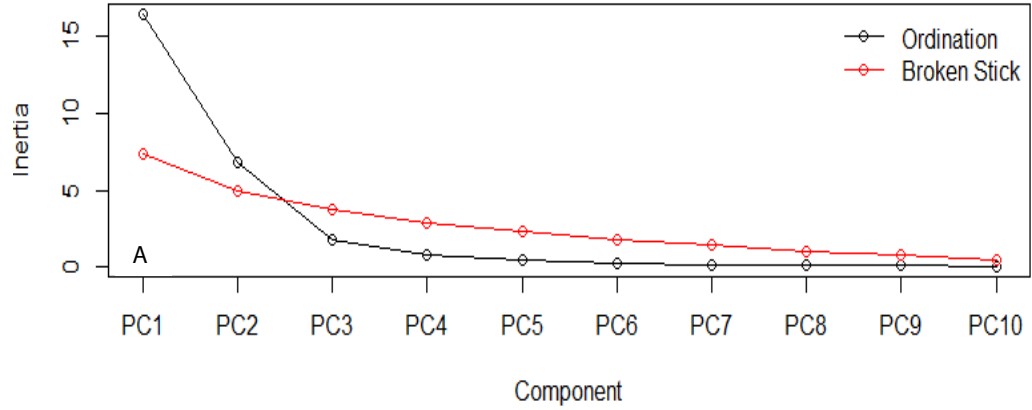

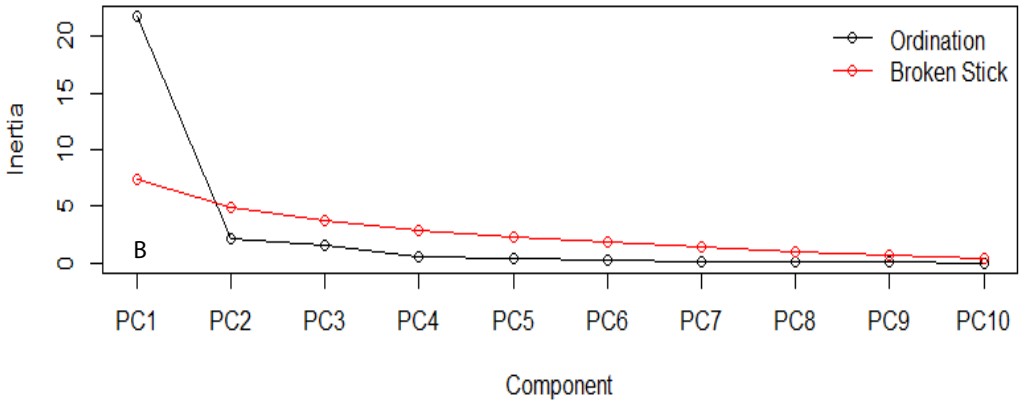

**Figure 5** **Broken stick distribution of the principal component between the ecological niche of transgenic PBR cowpea and its non-transgenic isoline.** (A) Transgenic cowpea; (B) Non-transgenic cowpea.

SC represents the most abundant species in the parasitoid guild of IT97KT and IT97KN ecological niches, while MB and AC are the most abundant species in the herbivore guild. CaC is the most abundant species in the predator guild. SL, LM, and vf represent the least abundant species in the predator, parasitoid, and herbivore guild of both ecological niches, as shown in Fig. 6. A uniform composition of the organisms in all the ecological niches was observed throughout the study period (Figs. 6A–6C).

## Dissimilarity index

The result of the Bray-Curtis dissimilarity index is presented in Table 3. The dissimilarity index between the ecological niches of PBR cowpea and non-transgenic isoline is 0.2, which indicates that all the niches had similar evolutionary trends with no divergence in the community structure of the non-targeted organisms.

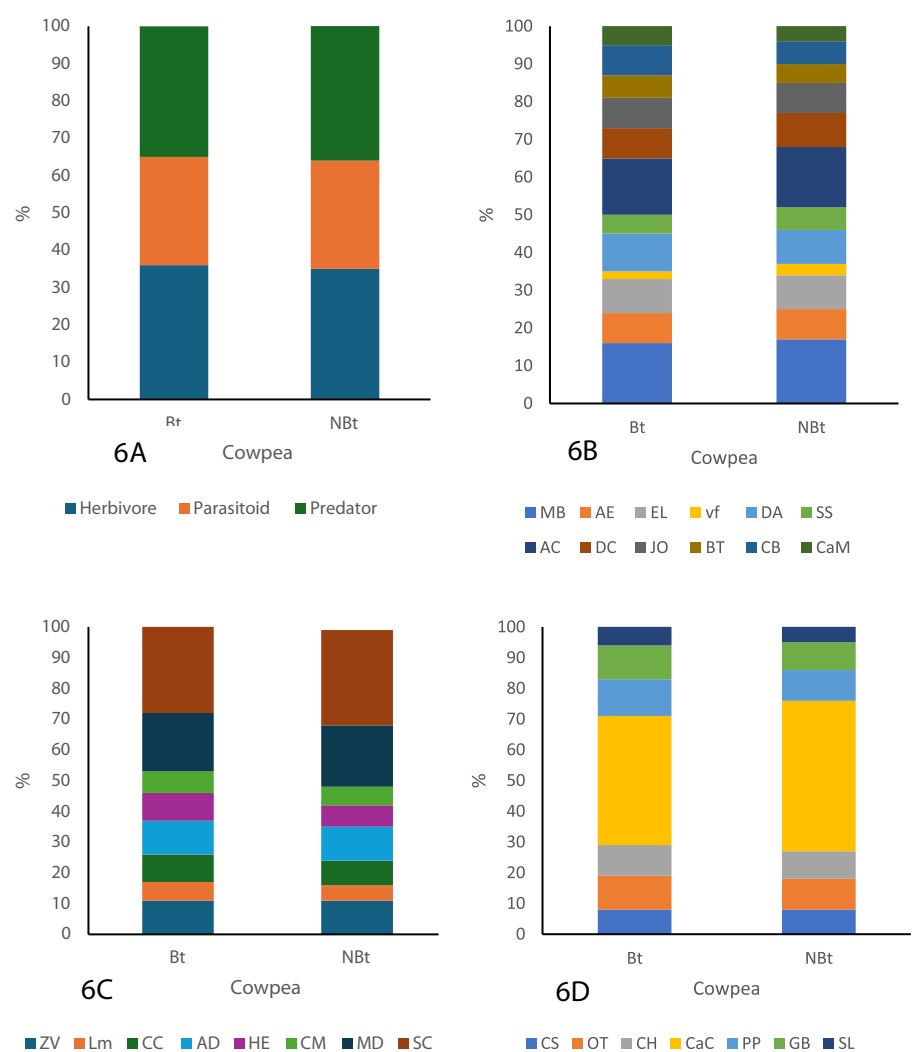

**Figure 6** Composition of the organism guild in both the *Bt* and N*Bt* fields.

**Table 3** Bray-Curtis dissimilarity index analysis.

| Descriptors | Values | Inference |
|---|---|---|
| $C_{ij}$ | 739 | |
| $S_i$ | 1,142 | 1. No divergence in NTOs community structure |
| $S_j$ | 739 | 2. Similar evolutionary trends |
| $BC_{ij}$ | 0.2 | |

**Notes.**

$C_{ij}$, the sum of the lesser values for the species found in each site; $S_i$, The total number of specimens counted at site I; $S_j$, The total number of specimens counted at site j; $BC_{ij}$, Bray-Curtis dissimilarity index.

## DISCUSSION

In this study, the potential impact of Nigeria's transgenic PBR cowpea, which is the first transgenic cowpea to be commercialized in the world, was assessed to evaluate the possible threats and harm that the crop may pose to the environment and the ecological niches of the diverse useful soil and plant organisms.

The current study observed a greater abundance of species and families across various ecological niches in transgenic PBR cowpea fields than in non-transgenic cowpea fields. This disparity may be attributed to the higher evenness and intensity of flowers in the IT97KT transgenic cowpea variety, leading to increased pod, leaf, and overall yield production. This speculation aligns with findings from several studies, including those by *Fragkiadaki et al. (2023)*, *Plos et al. (2023)*; *Bonelli et al. (2022)*; *Nadine et al. (2020)*; *Braatz et al. (2021)*, and *Adedoja, Kehinde & Samways (2018)*, all of which have linked flowering and podding to insect population dynamics.

According to *Guo et al. (2014)*, the various functional ecological indices of the surrounding species to any newly introduced crop such as the PBR cowpea would be significantly altered if disruption of any biological property occurs because of planting such crop. However, the findings from this research show that the total species count throughout the study period is similar in value. Analysis of the various ecological indices, including Shannon Diversity index, Bray-Curtis Dissimilarity index, Pielou evenness index, PCA, and Renyi Diversity silhouettes, all showed a close range of values between the ecological niches of the transgenic cowpea and its non-transgenic Isoline. A similar study conducted at Germany's Oderbruch European Corn Borer infestation area by *Schorling & Freier (2006)* on a six-year assessment of the impact of transgenic maize expressing Cry1Ab gene on non-target organisms reported the same results. In contrast to *Fernandes et al. (2022)*, who postulated that genetic modification of crops has the tendency to reduce crop biodiversity, findings by *Abdul et al. (2022)* and *Anderson et al. (2019)* indicated that the transformation of crops for insect resistance is beneficial because it can enable plant species that are near extinction because of the heavy burden of insect infestation to be revived by improving their adaptation to diverse environmental conditions. The findings from the current study unequivocally demonstrate that the incorporation of the Cry1Abgene into PBR cowpea does not adversely affect biodiversity.

The PCA of both the transgenic and non-transgenic cowpea fields reveals that the distribution of the NTOs was not significantly different throughout the study period. This finding is consistent with the report of *Guo et al. (2014)* and *Candolfi et al. (2009)*, who reported that the Cry1Ab event expressed in the transgenic Corn does not affect the community structure of the NTOs. Another research study by *Higgins et al. (2009)*, where a three-year field monitoring of the potential impacts of Cry1F events expressed in a maize hybrid on NTOs, also showed that the community structure of the organisms remained intact.

Though previous research only centred on the comparative NTO abundance between transgenic and non-transgenic plots, the present study further analyzed the possible evolutionary dynamics of the transgenic PBR cowpea by carrying out a dissimilarity index

analysis. The results show that there was a gradual change in the species composition of both transgenic fields and non-transgenic fields, and this change increased with time. For instance, the number of species present during CW 2 of the study increased compared to CW 1. A similar occurrence was also observed when CW 3 was compared with CW 2.

The Bray-Curtis Dissimilarity Index analysis showed an index of 0.2, suggesting that the evolutionary dynamic for transgenic and non-transgenic crops was significantly similar. Similar studies conducted by *Guo et al. (2014)* also recorded a similar evolutionary dynamic between non-transgenic and transgenic maize expressing *CryIAc* event. The potential toxicity of PBR cowpea can also be assessed by monitoring and evaluating the exposure of the various species' different life stages to cowpea in the ecosystem (*Devos et al., 2012*).

The assessment of the different nutritional guilds of organisms identified in this study shows a rich representation of the herbivores, parasitoids, and predators in all the ecological niches. Despite the high tendency of herbivores to have direct exposure to Cry proteins expressed in PBR cowpea when feeding on its crop residue and pollen (*Devos et al., 2012*; *Romeis et al., 2008*), a high population density was still recorded in the ecological niches of PBR cowpea compared to non-transgenic cowpea. The number of herbivore species present in the ecological niches of transgenic cowpea is higher than in the non-transgenic cowpea ecological niches but the same species type including *Messor barbarus* (Linnaeus, 1767), *Alydus eurinus* (Say, 1832); *Romalea microptera* (Beauvois, 1817), *Euptoieta claudia* (Cramer, 1775), *Deudorix antalus* (Hopffer, 1855), *Scarabaeus satyrus* (Fabricius, 1787), *Atta cephalotes* (Linnaeus, 1758), *Dysdercus cingulatus* (Fabricius, 1798), *Junonia oenone* (Linnaeus, 1758), *Chorthippus biguttulus* (Linnaeus, 1758) and *Carausius morosus* (Sinéty, 1901) were observed for all the ecological niches. This result is in line with findings from *Wolfenbarger et al. (2008)* who carried out a study on the potential impacts of transgenic crops on the functional guild of NTOs.

A further critical analysis of the population density of the predator guild in both transgenic and non-transgenic fields revealed no significant difference. Assessing the population density of the predator guild can provide valid assertions on the extent of biological, as well as environmental safety of the transgenic crop since predators have multiple ways by which they come in contact with the *Cry1Ab* gene, including direct feeding on the pollen of the PBR cowpea, herbivores that have feed on PBR cowpea or *via* the surrounding soil in which the PBR cowpea is planted.

The number of predator species present in the ecological niches of transgenic cowpea is higher than in the non-transgenic cowpea ecological niches though both had the same species type, including *Chilocorus stigma* (Say, 1832), *Odontoponera transversa* (Smith, 1858), *Conozoa hyaline* (Forbes, 1848), *Camponotus cruentatus* (Latreille, 1802), *Pirata piraticus* (Clerck,1757), *Graphoderus bilineatus* (De Geer, 1774) and *Stenolophus lecontei* (Chaudoir, 1869).

Analysis of the parasitoid population can provide some very useful ecological indices because they possess the unique characteristics of having the ability to complete their life cycle by feeding on a particular host (*Salama & Zaki, 1983*) or a range of herbivores in a particular ecological niche (*Romeis et al., 2008*). They are, therefore, most likely to ingest
the Cry protein in the host herbivore where they are found or directly from the PBR cowpea plant (*Lit et al., 2012*). The analysis shows that the population density of the parasitoids in the PBR cowpea ecological niches was not significantly different from the non-transgenic cowpea ecological niches throughout the study period. Research conducted by *Comas et al. (2014)* and *Albajes et al. (2013)*, who conducted a meta-analysis on the ecological impact of Bt Maize on non-target organisms (NTOs), similarly concluded that the transgenic maize did not exert a significant impact on the population density of predator, herbivore, and parasitoid guilds throughout the study.

The PCA result shows similar evolutionary dynamics in both the ecological niches of the transgenic and non-transgenic cowpea. The broken stick distribution, which models the number of variances by adopting a stick of unit length, which is thereafter randomly broken into n pieces, reveals no statistically significant difference between both ecological niches. This finding aligns with the result obtained by *Guo et al. (2014)*, whose research study revealed that the BtCry1Ac event expressed in the insect-resistant corn caused no alteration in the community distribution of both transgenic and non-transgenic corn.

The strong positive correlation between both transgenic and non-transgenic cowpea *vs* time shows that the increase in the species in both niches is a result of an increase in agronomic factors as the growth of both cowpea progresses. Such factors may include the onset of flowers and the steady increase, the onset of pods that followed thereafter, and its steady increase, in addition to the continuous increase in the number of leaves over time. It also means that the *Cry1Ab* gene expressed in the PBR cowpea had no negative impact on any of the ecological components, including the non-targeted organisms. Other factors that may have played significant roles include temperature, rainfall, sunshine, the nature of the soil, and other surrounding elements and plants (*Desneux & Bernal, 2010*).

The higher prevalence of species in transgenic PBR cowpea fields can be linked to multiple correlated factors, encompassing enhanced plant health and resource availability, specific interactions between the transgenic plants and their environment, disparities in nutritional content, and modified ecological interactions (*Yizhu et al., 2024*; *Bijay, Anju & Samikshya, 2023*; *Pandey, Vengavasi & Hawkesford, 2021*; *Zhe et al., 2010*): Transgenic PBR Cowpea is engineered to withstand attacks from pod borers, a significant pest in cowpea farming. With less harm inflicted by these pests, the transgenic plants could allocate more resources towards development and propagation, resulting in a potential rise in flower yield and enhanced nutritional value. This enhanced plant health might offer a more prosperous and superior supply of resources for various species, such as pollinators and herbivores. A research study by *Yizhu et al. (2024)* on core species impacting plant health by enhancing soil microbial cooperation and network complexity during community coalescence has further emphasized that healthy soil reduces the plant disease index and increases biomass by improving the stability and complexity of the network; positive cohesion, reflecting the degree of cooperation, was also negatively correlated with the plant disease index.

The presence of the *Cry1Ab* protein in transgenic PBR cowpea could directly or indirectly affect insect populations. *Cry1Ab* protein targets specific Lepidopteran pests, reducing their numbers and thus lessening the herbivory pressure on the plants. According to *Bijay, Anju & Samikshya (2023)*, reducing pest pressure could lead to a more favourable

environment for non-target insect species, as there would be less competition for resources and fewer damaged plants. The lower pest pressure might also reduce the need for chemical insecticides, further promoting a healthier ecosystem for a broader range of species.

Differences in the nutritional content of transgenic and non-transgenic cowpea plants could also play a role in the observed differences in species abundance (*Zhe et al., 2010*). Healthier, less stressed plants might produce higher levels of certain nutrients, attracting a more diverse array of herbivores and their predators (*Pandey, Vengavasi & Hawkesford, 2021*). This could create a cascading effect, supporting greater biodiversity in the transgenic PBR Cowpea fields. Moreover, these interactions could extend beyond herbivores to include pollinators and other beneficial insects, enhancing the overall ecological balance of the fields.

Introducing transgenic PBR cowpea could also alter the ecological interactions within the fields. For example, reducing pod borer populations might allow other species to thrive without the pressure of competition or predation from these pests. This could result in a more complex and diverse ecosystem where different species can exploit various niches. Additionally, the healthier plants might provide better habitat and resources for various organisms, from soil microbes to larger vertebrates, contributing to the observed increase in biodiversity.

A more in-depth study and analysis would contribute to substantiating the possible reasons for the observed differences in species counts. Some of these assessments may comprise detailed evaluations of insect populations, soil analyses, plant biochemical profiling, and the continuous monitoring of biodiversity throughout various growing seasons. Collaborating with ecologists, entomologists, and plant biologists can provide valuable insights and help elucidate the underlying mechanisms driving the observed patterns.

### Limitation of the current study

The current study does not consider the impact of PBR cowpea on the oviposition ability of non-targeted arthropods. Furthermore, the collection of data on the effect of PBR cowpea on soil invertebrates over longer periods of time and the potential transfer of the *Cry1Ab* gene to conventional cowpea still needs to be assessed.

## CONCLUSIONS

Data accrued from the analysis of the current study revealed no significant differences in the responses of non-targeted organisms between the ecological niches of the transgenic (IT97KT) and non-transgenic (IT97KN) cowpea. The findings from this study show that the introduction of the Cry1Abtransgene in the PBR cowpea did not negatively impact biodiversity and the environment. The comparative assessment of the evolutionary dynamics of the non-targeted species community of the transgenic cowpea and that of the non-transgenic cowpea recorded no significant divergence throughout the study period. The data accrued from the analysis of the species evenness and diversity indices also did not show any significant difference between the fields of transgenic PBR cowpea and its isoline. However, it is imperative to note that these findings are context-dependent

and may vary across different agroecosystems and geographical regions. Therefore, continuous monitoring and adaptive management strategies are essential to mitigate potential unforeseen consequences on biodiversity. This study found that the single-line transgenic cowpea (IT97KT) could thrive without or with reduced chemical pesticide usage, which, in turn, could lead to improved climate conditions and human health. However, it is important to take a cautious approach to minimize the risk of unintended ecological consequences, such as secondary pest outbreaks or disruption of natural enemy populations. The findings from this research provide valuable insights that will help shape decision-making for regulating the crop across all cowpea growing areas in the country.

### Funding
This work was supported by the African Agricultural Technology Foundation, Naivasha Rd, Nairobi, Kenya. The funders had no role in study design, data collection and analysis, decision to publish, or preparation of the manuscript.

### Grant Disclosures
The following grant information was disclosed by the authors:
African Agricultural Technology Foundation.

### Competing Interests
The authors declare there are no competing interests.

### Author Contributions
- Abraham Isah conceived and designed the experiments, performed the experiments, analyzed the data, prepared figures and/or tables, authored or reviewed drafts of the article, and approved the final draft.
- Rebeccah Wusa Ndana conceived and designed the experiments, performed the experiments, prepared figures and/or tables, authored or reviewed drafts of the article, and approved the final draft.
- Yoila David Malann conceived and designed the experiments, analyzed the data, prepared figures and/or tables, authored or reviewed drafts of the article, and approved the final draft.
- Onyekachi Francis Nwankwo conceived and designed the experiments, performed the experiments, analyzed the data, prepared figures and/or tables, authored or reviewed drafts of the article, and approved the final draft.
- Abdulrazak Baba Ibrahim conceived and designed the experiments, performed the experiments, analyzed the data, prepared figures and/or tables, and approved the final draft.
- Rose Suniso Gidado conceived and designed the experiments, performed the experiments, prepared figures and/or tables, authored or reviewed drafts of the article, and approved the final draft.

## Data Availability

The raw data are available in the Supplemental Files.

## Supplemental Information

Supplemental information for this article can be found online at http://dx.doi.org/10.7717/peerj.18094#supplemental-information.

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
