# Peer review of "Biodiversity assessment and environmental risk analysis of the single line transgenic pod borer resistant cowpea"

_PeerJ, doi:10.7717/peerj.18094_

## Round 0.1 · original submission · Major Revisions

Dear colleagues. Three reviewers have evaluated your manuscript and reported that it has merit for publication following a major revision. Please take into account the comments made also as annotated files. I would also suggest a more detailed temporal analysis (as proposed by the 2nd reviewer) since significant departures can occur across years in biological systems

**Language Note:** The review process has identified that the English language must be improved. PeerJ can provide language editing services - please contact us at [email protected] for pricing (be sure to provide your manuscript number and title). Alternatively, you should make your own arrangements to improve the language quality and provide details in your response letter. – PeerJ Staff

·

Basic reporting

The English form needs to be revised and improved. The rules of the International Biological Nomenclature must be followed: each species reported for the first timne in text should be written in full (in italics) with Authority and systematics.

Experimental design

The experimental part must be better detailed and better schematized. Above all what is referred to biodiversity indexes. And the arthropod systematics need to better detailed and explained

Validity of the findings

the results look interesting, need to be better explained above all in the discussion

·

Basic reporting

Dear authors, thank you for this important study. As it was clearly stated in the manuscript, it will help clear doubts about the so-called ‘’side effect’’ of Bt cowpea and increase the large adoption of the variety for food security in the region.
I read the manuscript with great interest.

The authors provide background about the study. I believe adding one or two sentences about a short background paragraph about ‘’Cry1Ab protein’’ about cowpea will help readers who are not familiar with the technology and improve the quality of the write-up.

On the format and English style used: Some of the sentences were very complex and this affected the flow of ideas. e.g lines 65-70, lines 94-98. In addition, there were many typos and mistakes of grammar throughout. For instance, the word ‘’Species’’ should always be with ‘’s’’ not ‘’specie’’.

The article is well structured; however, all figures (Figures s1, 4, 6) should stand alone. Kindly provide a clear and complete legend.

‘’All species and families observed during this study were more abundant in transgenic PBR Cowpea fields than in non-transgenic cowpea fields’’. You failed to clearly explain this and the possible reasons behind it. What factor could have contributed to the reduction observed in the non-transgenic field If no insecticide was applied???

Experimental design

The authors submitted an original manuscript, and I believe the topic discussed therein aligns well with the Aims and Scope of the journal. The research question is well-defined, relevant & meaningful.
The methodological approach used was described and the experiment was replicated over time.
There is a need to adopt a clear and uniform abbreviation for the transgenic and non-transgenic lines. The use of multiple abbreviations makes it very hard to read. See lines 117, 122 (IT97KT, IT97K, ITN7KN), Figure 4 (Bt, Non Bt)

Validity of the findings

All underlying data have been provided. The significance threshold for the validity of the results should be indicated in the methodology to avoid confusion. For instance, in lines 209 -2012 you have declared some comparisons between treatments are not significant yet the probability values are below 5%.
Unless otherwise understood, the study was carried out across two seasons and you should present and interpret both the individual season as well as across season variations.
The conclusions are stated and linked to the original research question.

Additional comments

Kindly you need proper editing of the English language throughout the manuscript. The discussion is very shallow and should be improved to better draw the attention of the readers beyond the values presented in the study.

Reviewer 3 ·

Basic reporting

The English language should be improved to ensure that an international audience can clearly understand your text. Some examples where the language could be improved include
Line 85 suggest “potential detrimental”
Line 92 change “the general debate In Africa on the potential impact of GM crops on biodiversity has triggered” ” a general debate in Africa on the potential impact of GM crops on biodiversity has been triggered”
Line 95 Change , “More apparent among” to “Among”
Line 108, Make it clear IT97KT is aka SAMPEA 20-T but then continue to use IT97KT
Line 113 mortals??
Line 118 What is IT97KY?
Line 123 ITN7KN??

Experimental design

Meets all the criteria.

Validity of the findings

The findings seem valid to me.

Additional comments

I suggest that a colleague who is proficient in English and familiar with the subject matter review the manuscript, or use a professional editing service.

---

## Round 0.2 · Major Revisions

Dear colleagues. Two reviewers have now completed their assessment of your manuscript. The manuscript has been substantially improved. still, one of them (and I also agree) requires a more analytical description based on seasonal and across-season variations when using transgenic lines. This would provide a better view of how environmental drivers can interact and weigh (if) the outcome. Hence, my decision is a major revision

·

Basic reporting

The authors did a great job adding some information on Cry1Ab and PBR protein in cowpea. To further improve the write-up's quality, I suggest they include references for this information (lines 137 to 199). Kindly also add Bt (Bacillus thuringiensis), to make the abbreviation clear to readers. Additionally, I suggest you add references for the new sentences (lines 216 to 220).

‘’All species and families observed during this study were more abundant in transgenic PBR Cowpea fields than in non-transgenic cowpea fields’’. While your explanation and comparison with previous studies provide some context, it would improve the write-up if you delve deeper into the underlying ecological or biological mechanisms that might explain the observed differences. It is not very convincing to attribute the greater abundance to flower intensity and production only. You can consider discussing factors such as resource availability, plant health, or specific interactions between the transgenic plants and their environment. For instance, how might the presence of the Cry1Ab protein affect insect populations directly or indirectly? Could there be differences in nutritional content, or other ecological interactions that are influencing these dynamics? Providing a more comprehensive explanation would help readers understand the full scope of your findings beyond the comparison with previous studies.
Please cite the author of the lme4 package (line 445).

Does the result title (line 851) cover all the subsequent subtitles or just the subtitle at line 852?

Kindly ask colleagues to proofread the manuscript for English language editing.

Experimental design

Nothing to add

Validity of the findings

Thank you for the argument about the seasonal variation. You highlighted that the study was designed to provide an overall assessment of the effects of transgenic lines on biodiversity and environmental factors, and for that, you decided to present the across seasons data only. This is arguable and very shallow. Assessing an overall impact can prevent one from appreciating specific trends in the data and the readers/users of your research to achieve unbiased decision-making. Please, if you cannot address this comment perhaps you should add data of the seasonal variations in the supplementary files.

Remember that we are already facing public opinions about GMOs and I believe that is why your research is very important. Hence, we should endeavor to spell out and present specific evidence to support our claims rather than drawing conclusions based on general/overall analyses. You should therefore provide more in-depth analyses. Even researchers are part of the stakeholders. So, if you want to convince these people provide strong evidence that your conclusions are valid.

Additional comments

Nothing to add

Reviewer 3 ·

Basic reporting

This ms now meets my expectation of language clarity.

Experimental design

Well described

Validity of the findings

I judge them to be accepted

Additional comments

I wonder why they do not describe the source of the seed with a peer-reviewed reference: doi: 10.1093/jee/toz367. This unfortunately means that I am requesting a citation of a paper on which I am a minor author.

---

## Round 0.3 · accepted · Accept

Dear colleague, your revisions have been processed and I believe that your manuscript is suitable for publication as it stands

·

Basic reporting

The authors have addressed the requested revisions. The manuscript has been improved. I have noted a few minor adjustments to the reference style directly in the main text. Kindly proofread the manuscript using a language editing service or someone with a good command of English.

Experimental design

No comment

Validity of the findings

No comment

Additional comments

No comment